# RETHINKING THE ROLE OF GRADIENT-BASED ATTRIBUTION METHODS FOR MODEL INTERPRETABILITY

**Suraj Srinivas**
Idiap Research Institute & EPFL
suraj.srinivas@idiap.ch

**François Fleuret**
University of Geneva
francois.fleuret@unige.ch

## ABSTRACT

Current methods for the interpretability of discriminative deep neural networks commonly rely on the model's input-gradients, i.e., the gradients of the output logits w.r.t. the inputs. The common assumption is that these input-gradients contain information regarding $p_\theta(y \mid \mathbf{x})$, the model's discriminative capabilities, thus justifying their use for interpretability. However, in this work we show that these input-gradients can be arbitrarily manipulated as a consequence of the shift-invariance of softmax without changing the discriminative function. This leaves an open question: if input-gradients can be arbitrary, why are they highly structured and explanatory in standard models?

We investigate this by re-interpreting the logits of standard softmax-based classifiers as unnormalized log-densities of the data distribution and show that input-gradients can be viewed as gradients of a class-conditional density model $p_\theta(\mathbf{x} \mid y)$ implicit within the discriminative model. This leads us to hypothesize that the highly structured and explanatory nature of input-gradients may be due to the alignment of this class-conditional model $p_\theta(\mathbf{x} \mid y)$ with that of the ground truth data distribution $p_{\text{data}}(\mathbf{x} \mid y)$. We test this hypothesis by studying the effect of density alignment on gradient explanations. To achieve this density alignment, we use an algorithm called score-matching, and propose novel approximations to this algorithm to enable training large-scale models.

Our experiments show that improving the alignment of the implicit density model with the data distribution enhances gradient structure and explanatory power while reducing this alignment has the opposite effect. This also leads us to conjecture that unintended density alignment in standard neural network training may explain the highly structured nature of input-gradients observed in practice. Overall, our finding that input-gradients capture information regarding an implicit generative model implies that we need to re-think their use for interpreting discriminative models.

## 1 INTRODUCTION

Input-gradients, or gradients of outputs w.r.t. inputs, are commonly used for the interpretation of deep neural networks (Simonyan et al., 2013). For image classification tasks, an input pixel with a larger input-gradient magnitude is attributed a higher 'importance' value, and the resulting maps are observed to agree with human intuition regarding which input pixels are important for the task at hand (Adebayo et al., 2018). Quantitative studies (Samek et al., 2016; Shrikumar et al., 2017) also show that these importance estimates are meaningful in predicting model response to larger structured perturbations. These results suggest that input-gradients do indeed capture relevant information regarding the underlying model. However in this work, we show that input-gradients can be arbitrarily manipulated using the shift-invariance of softmax without changing the underlying discriminative model, which calls into question the reliability of input-gradient based attribution methods for interpreting arbitrary black-box models.

Given that input-gradients can be arbitrarily structured, the reason for their highly structured and explanatory nature in standard pre-trained models is puzzling. Why are input-gradients relatively well-

behaved when they can just as easily be arbitrarily structured, without affecting discriminative model performance? What factors influence input-gradient structure in standard deep neural networks?

To answer these, we consider the connections made between softmax-based discriminative classifiers and generative models (Bridle, 1990; Grathwohl et al., 2020), made by viewing the logits of standard classifiers as un-normalized log-densities. This connection reveals an alternate interpretation of input-gradients, as representing the log-gradients of a class-conditional density model which is implicit within standard softmax-based deep models, which we shall call the *implicit density model*. This connection compels us to consider the following hypothesis: perhaps input-gradients are highly structured because this implicit density model is aligned with the 'ground truth' class-conditional data distribution? The core of this paper is dedicated to testing the validity of this hypothesis, whether or not input-gradients do become more structured and explanatory if this alignment increases and vice versa.

For the purpose of validating this hypothesis, we require mechanisms to increase or decrease the alignment between the implicit density model and the data distribution. To this end, we consider a generative modelling approach called score-matching, which reduces the density modelling problem to that of local geometric regularization. Hence by using score-matching, we are able to view commonly used geometric regularizers in deep learning as density modelling methods. In practice, the score-matching objective is known for being computationally expensive and unstable to train (Song & Ermon, 2019; Kingma & LeCun, 2010). To this end, we also introduce approximations and regularizers which allow us to use score-matching on practical large-scale discriminative models.

This work is broadly connected to the literature around unreliability of saliency methods. While most such works consider how the explanations for nearly identical images can be arbitrarily different (Dombrowski et al., 2019; Subramanya et al., 2019; Zhang et al., 2020; Ghorbani et al., 2019), our work considers how one may change the model itself to yield arbitrary explanations without affecting discriminative performance. This is similar to Heo et al. (2019) who show this experimentally, whereas we provide an analytical reason for why this happens relating to the shift-invariance of softmax.

The rest of the paper is organized as follows. We show in § 2 that it is trivial to manipulate input-gradients of standard classifiers using the shift-invariance of softmax without affecting the discriminative model. In § 3 we state our main hypothesis and describe the details of score-matching, present a tractable approximation for the same that eliminates the need for expensive Hessian computations. § 4 revisits other interpretability tools from a density modelling perspective. Finally, § 5 presents experimental evidence for the validity of the hypothesis that improved alignment between the implicit density model and the data distribution can improve the structure and explanatory nature of input-gradients.

## 2    INPUT-GRADIENTS ARE NOT UNIQUE

In this section, we show that it is trivial to manipulate input-gradients of discriminative deep networks, using the well-known shift-invariance property of softmax. Here we shall make a distinction between two types of input-gradients: *logit-gradients* and *loss-gradients*. While logit-gradients are gradients of the pre-softmax output of a given class w.r.t. the input, loss-gradients are the gradients of the loss w.r.t. the input. In both cases, we only consider outputs of a single class, usually the target class.

Let $\mathbf{x} \in \mathbb{R}^D$ be a data point, which is the input for a neural network model $f : \mathbb{R}^D \rightarrow \mathbb{R}^C$ intended for classification, which produces pre-softmax logits for $C$ classes. The cross-entropy loss function for some class $1 \leq i \leq C$, $i \in \mathbb{N}$ corresponding to an input $\mathbf{x}$ is given by $\ell(f(\mathbf{x}), i) \in \mathbb{R}_+$, which is shortened to $\ell_i(\mathbf{x})$ for convenience. Note that here the loss function subsumes the softmax function as well. The logit-gradients are given by $\nabla_{\mathbf{x}} f_i(\mathbf{x}) \in \mathbb{R}^D$ for class $i$, while loss-gradients are $\nabla_{\mathbf{x}} \ell_i(\mathbf{x}) \in \mathbb{R}^D$. Let the softmax function be $p(y = i | \mathbf{x}) = \exp(f_i(\mathbf{x})) / \sum_{j=1}^{C} \exp(f_j(\mathbf{x}))$, which we denote as $p_i$ for simplicity. Here, we make the observation that upon adding the same scalar function $g$ to all logits, the logit-gradients can arbitrarily change but the loss values do not.

**Observation.** *Assume an arbitrary function $g : \mathbb{R}^D \rightarrow \mathbb{R}$. Consider another neural network function given by $\tilde{f}_i(\cdot) = f_i(\cdot) + g(\cdot)$, for $0 \leq i \leq C$, for which we obtain $\nabla_{\mathbf{x}} \tilde{f}_i(\cdot) = \nabla_{\mathbf{x}} f_i(\cdot) + \nabla_{\mathbf{x}} g(\cdot)$.*

*For this, the corresponding loss values and loss-gradients are unchanged, i.e.; $\tilde{\ell}_i(\cdot) = \ell_i(\cdot)$ and $\nabla_{\mathbf{x}} \tilde{\ell}_i(\cdot) = \nabla_{\mathbf{x}} \ell_i(\cdot)$ as a consequence of the shift-invariance of softmax.*

This explains how the structure of logit-gradients can be arbitrarily changed: one simply needs to add an arbitrary function $g$ to all logits. This implies that individual logit-gradients $\nabla_{\mathbf{x}} f_i(\mathbf{x})$ and logits $f_i(\mathbf{x})$ are meaningless on their own, and their structure may be uninformative regarding the underlying discriminative model. Despite this, a large fraction of work in interpretable deep learning (Simonyan et al., 2013; Selvaraju et al., 2017; Smilkov et al., 2017; Fong et al., 2019; Srinivas & Fleuret, 2019) uses individual logits and logit-gradients for saliency map computation. We also provide a similar illustration in the supplementary material for the case of loss-gradients, where we show that it is possible for loss-gradients to diverge significantly even when the loss values themselves do not.

These simple observations leave an open question: why are input-gradients highly structured and explanatory when they can just as easily be arbitrarily structured, without affecting discriminative model performance? Further, if input-gradients do not depend strongly on the underlying discriminative function, what aspect of the model do they depend on instead? In the section that follows, we shall consider a generative modelling view of discriminative neural networks that offers insight into the information encoded by logit-gradients.

## 3 IMPLICIT DENSITY MODELS WITHIN DISCRIMINATIVE CLASSIFIERS

Let us consider the following link between generative models and the softmax function. We first define the following joint density on the logits $f_i$ of classifiers: $p_\theta(\mathbf{x}, y = i) = \frac{\exp(f_i(\mathbf{x};\theta))}{Z(\theta)}$, where $Z(\theta)$ is the partition function. We shall henceforth suppress the dependence of $f$ on $\theta$ for brevity. Upon using Bayes' rule to obtain $p_\theta(y = i \mid \mathbf{x})$, we observe that we recover the standard softmax function. Thus the logits of discriminative classifiers can alternately be viewed as un-normalized log-densities of the joint distribution. Assuming equiprobable classes, we have $p_\theta(\mathbf{x} \mid y = i) = \frac{\exp(f_i(\mathbf{x}))}{Z(\theta)/C}$, which is the quantity of interest for us. Thus while the logits represent un-normalized log-densities, logit-gradients represent the score function, i.e.; $\nabla_x \log p_\theta(\mathbf{x} \mid y = i) = \nabla_x f_i(\mathbf{x})$, which avoids dependence on the partition function $Z(\theta)$ as it is independent of $\mathbf{x}$.

This viewpoint naturally leads to the following hypothesis, that perhaps the reason for the highly structured and explanatory nature of input-gradients is that the implicit density model $p_\theta(\mathbf{x} \mid y)$ is close to that of the ground truth class-conditional data distribution $p_{\text{data}}(\mathbf{x} \mid y)$? We propose to test this hypothesis explicitly using score-matching as a density modelling tool.

**Hypothesis.** *(Informal) Improved alignment of the implicit density model to the ground truth class-conditional density model improves input-gradient interpretability via both qualitative and quantitative measures, whereas deteriorating this alignment has the opposite effect.*

### 3.1 SCORE-MATCHING

Score-matching (Hyvärinen, 2005) is a generative modelling objective that focusses solely on the derivatives of the log density instead of the density itself, and thus does not require access to the partition function $Z(\theta)$. Specifically, for our case we have $\nabla_{\mathbf{x}} \log p_\theta(\mathbf{x} \mid y = i) = \nabla_{\mathbf{x}} f_i(\mathbf{x})$, which are the logit-gradients.

Given i.i.d. samples $\mathcal{X} = \{x_i \in \mathbb{R}^D\}$ from a latent data distribution $p_{data}(\mathbf{x})$, the objective of generative modelling is to recover this latent distribution using only samples $\mathcal{X}$. This is often done by training a parameterized distribution $p_\theta(\mathbf{x})$ to align with the latent data distribution $p_{data}(\mathbf{x})$. The score-matching objective instead aligns the gradients of log densities, as given below.

$$
\begin{aligned}
J(\theta) &= \mathbb{E}_{p_{data}(\mathbf{x})} \frac{1}{2} \|\nabla_{\mathbf{x}} \log p_\theta(\mathbf{x}) - \nabla_{\mathbf{x}} \log p_{data}(\mathbf{x})\|_2^2 && (1) \\
&= \mathbb{E}_{p_{data}(\mathbf{x})} \left( \text{trace}(\nabla_{\mathbf{x}}^2 \log p_\theta(\mathbf{x})) + \frac{1}{2} \|\nabla_{\mathbf{x}} \log p_\theta(\mathbf{x})\|_2^2 \right) + \texttt{const} && (2)
\end{aligned}
$$

The above relationship is proved (Hyvärinen, 2005) using integration by parts. This is a consistent objective, *i.e.,* $J(\theta) = 0 \iff p_{data} = p_\theta$. This approch is appealing also because this reduces the problem of generative modelling to that of regularization of the local geometry of functions, i.e.; the resulting terms only depend on the point-wise gradients and Hessian-trace.

## 3.2 Efficient estimation of Hessian-trace

In general, equation 2 is intractable for high-dimensional data due to the Hessian trace term. To address this, we can use the Hutchinson's trace estimator (Hutchinson, 1990) to efficiently compute an estimate of the trace by using random projections, which is given by: $\text{trace}(\nabla_{\mathbf{x}}^2 \log p_\theta(\mathbf{x})) = \mathbb{E}_{\boldsymbol{v} \sim \mathcal{N}(0,\mathbf{I})} \boldsymbol{v}^{\mathsf{T}} \nabla_{\mathbf{x}}^2 \log p_\theta(\mathbf{x}) \, \boldsymbol{v}$. This estimator has been previously applied to score-matching (Song et al., 2019), and can be computed efficiently using Pearlmutter's trick (Pearlmutter, 1994). However, this trick still requires **two backward passes** for a single monte-carlo sample, which is computationally expensive. To further improve computational efficiency, we introduce the following approximation to Hutchinson's estimator using a Taylor series expansion, which applies to small values of $\sigma \in \mathbb{R}$.

$$
\begin{aligned}
\mathbb{E}_{\boldsymbol{v} \sim \mathcal{N}(0,\mathbf{I})} \boldsymbol{v}^{\mathsf{T}} \nabla_{\mathbf{x}}^2 \log p_\theta(\mathbf{x}) \boldsymbol{v} \quad &\approx \quad \frac{2}{\sigma^2} \mathbb{E}_{\boldsymbol{v} \sim \mathcal{N}(0,\sigma^2\mathbf{I})} \left( \log p_\theta(\mathbf{x} + \boldsymbol{v}) - \log p_\theta(\mathbf{x}) - \nabla_x \log p_\theta(\mathbf{x})^{\mathsf{T}} \boldsymbol{v} \right) \\
&= \quad \frac{2}{\sigma^2} \mathbb{E}_{\boldsymbol{v} \sim \mathcal{N}(0,\sigma^2\mathbf{I})} \left( \log p_\theta(\mathbf{x} + \boldsymbol{v}) - \log p_\theta(\mathbf{x}) \right) \quad\quad (3)
\end{aligned}
$$

Note that equation 3 involves a difference of log probabilities, which is independent of the partition function. For our case, $\log p_\theta(\mathbf{x} + \boldsymbol{v} | y = i) - \log p_\theta(\mathbf{x} | y = i) = f_i(\mathbf{x} + \boldsymbol{v}) - f_i(\mathbf{x})$. We have thus considerably simplified and speeded-up the computation of the Hessian trace term, which now can be approximated with **no backward passes**, but using only a single additional forward pass. We present details regarding the variance of this estimator in the supplementary material. A concurrent approach (Pang et al., 2020) also presents a similar algorithm, however it is applied primarily to Noise Contrastive Score Networks (Song & Ermon, 2019) and Denoising Score Matching (Vincent, 2011), whereas we apply it to vanilla score-matching on discriminative models.

## 3.3 Stabilized Score-matching

In practice, a naive application of score-matching objective is unstable, causing the Hessian-trace to collapse to negative infinity. This occurs because the finite-sample variant of equation 1 causes the model to 'overfit' to a mixture-of-diracs density, which places a dirac-delta distribution at every data point. Gradients of such a distribution are undefined, causing training to collapse. To overcome this, regularized score-matching (Kingma & LeCun, 2010) and noise conditional score networks (Song & Ermon, 2019) propose to add noise to inputs for score-matching to make the problem well-defined. However, this did not help for our case. Instead, we use a heuristic where we add a small penalty term proportional to the square of the Hessian-trace. This discourages the Hessian-trace becoming too large, and thus stabilizes training.

## 4 Implications of the Density Modelling Viewpoint

In the previous section we related input-gradients to the implicit density model, thus linking gradient interpretability to density modelling through our hypothesis. In this section, we consider two other interpretability tools: activity maximization and the pixel perturbation test, and show how these can interpreted from a density modelling perspective. These perspectives also enable us to draw parallels between score-matching and adversarial training.

### 4.1 Activity Maximization as Sampling from the Implicit Density Model

The canonical method to obtain samples from score-based generative models is via Langevin sampling (Welling & Teh, 2011; Song & Ermon, 2019), which involves performing gradient ascent on the density model with noise added to the gradients. Without this added noise, the algorithm recovers the modes of the density model.

We observe that activity maximization algorithms used for neural network visualizations are remarkably similar to this scheme. For instance, Simonyan et al. (2013) recover inputs which maximize the logits of neural networks, thus exactly recovering the modes of the implicit density model. Similarly, deep-dream-like methods (Mahendran & Vedaldi, 2016; Nguyen et al., 2016; Mordvintsev et al., 2015) extend this by using "image priors" to ensure that the resulting samples are closer to the distribution of natural images, and by adding structured noise to the gradients in the form of jitter, to obtain more visually pleasing samples. From the density modelling perspective, we can alternately view these visualization techniques as biased sampling methods for score-based density models trained on natural images. However, given the fact that they draw samples from the implicit density model, their utility in interpreting discriminative models may be limited.

## 4.2 PIXEL PERTURBATION AS A DENSITY RATIO TEST

A popular test for saliency map evaluation is based on pixel perturbation (Samek et al., 2016). This involves first selecting the least-relevant (or most-relevant) pixels according to a saliency map representation, 'deleting' those pixels and measuring the resulting change in output value. Here, deleting a pixel usually involves replacing the pixel with a non-informative value such as a random or a fixed constant value. A good saliency method identifies those pixels as less relevant whose deletion does not cause a large change in output value.

We observe that this change in outputs criterion is identical to the density ratio, *i.e.*, $\log\left(p_\theta(\mathbf{x} + \boldsymbol{v}|y = i)/p_\theta(\mathbf{x}|y = i)\right) = f_i(\mathbf{x} + \boldsymbol{v}) - f_i(\mathbf{x})$. Thus when logits are used for evaluating the change in outputs (Samek et al., 2016; Ancona et al., 2018), the pixel perturbation test exactly measures the density ratio between the perturbed image and the original image. Thus if a perturbed image has a similar density to that of the original image under the implicit density model, then the saliency method that generated these perturbations is considered to be explanatory. Similarly, Fong et al. (2019) optimize over this criterion to identify pixels whose removal causes minimal change in logit activity, thus obtaining perturbed images with a high implicit density value similar to that of activity maximization. Overall, this test captures sensitivity of the implicit density model, and not the underlying discriminative model which we wish to interpret. We thus recommend that the pixel perturbation test always be used in conjunction with either the change in output probabilities, or the change in the accuracy of classification, rather than the change in logits.

## 4.3 CONNECTING SCORE-MATCHING TO ADVERSARIAL TRAINING

Recent works in adversarial machine learning (Etmann et al., 2019; Engstrom et al., 2019; Santurkar et al., 2019; Kaur et al., 2019; Ross & Doshi-Velez, 2017) have observed that saliency map structure and samples from activation maximization are more perceptually aligned for adversarially trained models than for standard models. However it is unclear from these works why this occurs. Separate from this line of work, Chalasani et al. (2018) also connect regularization of a variant of integrated gradients with adversarial training, suggesting a close interplay between the two.

We notice that these properties are shared with score-matched models, or models trained such that the implicit density model is aligned with the ground truth. Further, we note that both score-matching and adversarial training are often based on local geometric regularization, usually involving regularization of the gradient-norm (Ross & Doshi-Velez, 2017; Jakubovitz & Giryes, 2018), and training both the discriminative model and the implicit density model (Grathwohl et al., 2020) has been shown to improve adversarial robustness. From these results, we can conjecture that training the implicit density model via score-matching may have similar outcomes as adversarial training. We leave the verification and proof of this conjecture to future work.

## 5 EXPERIMENTS

In this section, we present experimental results to show the efficacy of score-matching and the validation of the hypothesis that density alignment influences the gradient explanation quality. For experiments, we shall consider the CIFAR100 dataset. We present experiments with CIFAR10 in the supplementary section. Unless stated otherwise, the network structure we use shall be a 18-layer ResNet that achieves 78.01% accuracy on CIFAR100, and the optimizer used shall be SGD with momentum. All models use the softplus non-linearity with $\beta = 10$, which is necessary to ensure that

the Hessian is non-zero for score-matching. Before proceeding with our experiments, we shall briefly introduce the score-matching variants we shall be using for comparisons.

**Score-Matching** We propose to use the score-matching objective as a regularizer in neural network training to **increase** the alignment of the implicit density model to the ground truth, as shown in equation 4, with the stability regularizer discussed in §3.3. For this, we use a regularization constant $\lambda = 1e - 3$. This model achieves 72.20% accuracy on the test set, which is a drop of about 5.8% compared to the original model. In the supplementary material, we perform a thorough hyper-parameter sweep and show that it is possible to obtain better performing models.

$$h(\mathbf{x}) := \frac{2}{\sigma^2} \mathbb{E}_{\boldsymbol{v} \sim \mathcal{N}(0, \sigma^2 \mathbf{I})} \left( f_i(\mathbf{x} + \boldsymbol{v}) - f_i(\mathbf{x}) \right)$$

$$\underbrace{\ell_{reg}(f(\mathbf{x}), i)}_{\text{regularized loss}} = \underbrace{\ell(f(\mathbf{x}), i)}_{\text{cross-entropy}} + \lambda \left( \underbrace{\overbrace{h(\mathbf{x})}^{\text{Hessian-trace}} + \frac{1}{2} \overbrace{\|\nabla_{\mathbf{x}} f_i(\mathbf{x})\|_2^2}^{\text{gradient-norm}}}_{\text{score-matching}} + \underbrace{\overbrace{\mu}^{10^{-4}} h^2(\mathbf{x})}_{\text{stability regularizer}} \right) \tag{4}$$

**Anti-score-matching** We would like to have a tool that can **decrease** the alignment between the implicit density model and the ground truth. To enable this, we propose to maximize the hessian-trace, in an objective we call *anti-score-matching*. For this, we shall use a the clamping function on hessian-trace, which ensures that its maximization stops after a threshold is reached. We use a threshold of $\tau = 1000$, and regularization constant $\lambda = 1e - 4$. This model achieves an accuracy of 74.87%.

**Gradient-Norm regularization** We propose to use gradient-norm regularized models as another baseline for comparison, using a regularization constant of $\lambda = 1e - 3$. This model achieves an accuracy of 76.60%.

### 5.1 EVALUATING THE EFFICACY OF SCORE-MATCHING AND ANTI-SCORE-MATCHING

Here we demonstrate that training with score-matching / anti-score-matching is possible, and that such training improves / deteriorates the quality of the implicit density models respectively as expected.

#### 5.1.1 DENSITY RATIOS

One way to characterize the generative behaviour of models is to compute likelihoods on data points. However this is intractable for high-dimensional problems, especially for un-normalized models. We observe although that the densities $p(\mathbf{x} \mid y = i)$ themselves are intractable, we can easily compute density ratios $p(\mathbf{x} + \eta \mid y = i)/p(\mathbf{x} \mid y = i) = \exp(f_i(\mathbf{x} + \eta) - f_i(\mathbf{x}))$ for a random noise variable $\eta$. Thus, we propose to plot the graph of density ratios locally along random directions. These can be thought of as local cross-sections of the density sliced at random directions. We plot these values for gaussian noise $\eta$ for different standard deviations, which are averaged across points in the entire dataset.

In Figure 1, we plot the density ratios upon training on the CIFAR100 dataset. We observe that the baseline model assigns *higher* density values to noisy inputs than real inputs. With anti-score-matching, we observe that the density profile grows still steeper, assigning higher densities to inputs with smaller noise. Gradient-norm regularized models and score-matched models improve on this behaviour, and are robust to larger amounts of noise added. Thus we are able to obtain penalty terms that can both improve and deteriorate the density modelling behaviour within discriminative models.

#### 5.1.2 SAMPLE QUALITY

We are interested in recovering modes of our density models while having access to only the gradients of the log density. For this purpose, we apply gradient ascent on the log probability $\log p(\mathbf{x} \mid y = i) = f_i(\mathbf{x})$, similar to activity maximization. Our results are shown in Figure 2. We observe that samples from the score-matched and gradient-norm regularized models are significantly less noisy than other models.

| Model | GAN-test (%) |
|---|---|
| Baseline ResNet | 59.47 |
| + Anti-Score-Matching | 16.40 |
| + Gradient Norm-regularization | **80.07** |
| + Score-Matching | 72.75 |

Table 1: GAN-test scores (higher is better) of class-conditional samples generated from various ResNet-18 models (see § 5.1.2). We observe that samples from gradient-norm regularized models and score-matched models achieve much better accuracies than the baselines and anti-score-matched models.

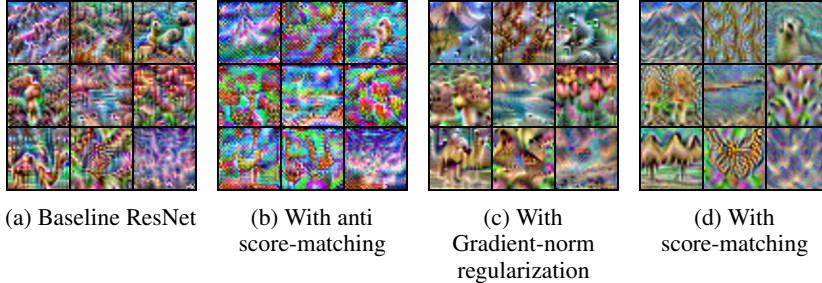

(a) Baseline ResNet    (b) With anti score-matching    (c) With Gradient-norm regularization    (d) With score-matching

Figure 2: Samples generated from various models by performing gradient ascent on random inputs (see § -5.1.2). While none of the generated samples are realistic, samples obtained from score-matched and gradient-norm regularized models are smoother and less noisy.

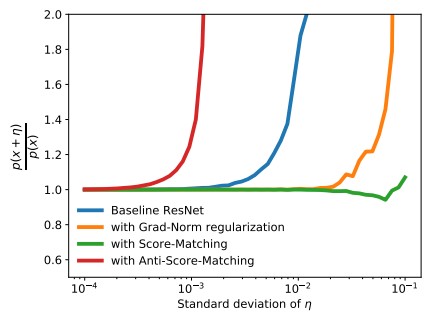

Figure 1: Plots of density ratios representing local density profiles across varying levels of noise added to the input (lower is better). We observe that score-matched model is robust to a larger range of noise values, while anti-score-matching is very sensitive even to small amounts of noise.

We also propose to qualitatively measure the sample quality using the GAN-test approach (Shmelkov et al., 2018). This test proposes to measure the discriminative accuracy of generated samples via an independently trained discriminative model. In contrast with more popular metrics such as the inception-score, this captures sample quality rather than diversity, which is what we are interested in. We show the results in table 1, which confirms the qualitative trend seen in samples above. Surprisingly, we find that gradient-norm regularized models perform better than score-matched models. This implies that such models are able to implicitly perform density modelling without being explicitly trained to do so. We leave further investigation of this phenomenon to future work.

## 5.2 EVALUATING THE EFFECT OF DENSITY ALIGNMENT ON GRADIENT EXPLANATIONS

Here we shall evaluate the gradient explanations of various models. First, we shall look at quantitative results on a discriminative variant of the pixel perturbation test. Second, we visualize the gradient maps to assess qualitative differences between them.

### 5.2.1 QUANTITATIVE RESULTS ON DISCRIMINATIVE PIXEL PERTURBATION

As noted in 4.2, it is recommended to use the pixel perturbation test using accuracy changes, and we call this variant as *discriminative pixel perturbation*. We select the least relevant pixels and replace them with the mean pixel value of the image, note down the accuracy of the model on the resulting samples. We note that this test is only used so far to compare different saliency methods for the same underlying model. However, we here seek to compare saliency methods across models. For this we consider two experiments. First, we perform the pixel perturbation experiment with each of the four

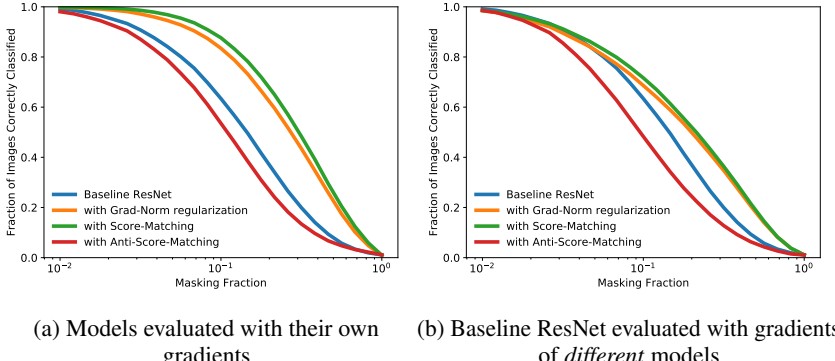

(a) Models evaluated with their own gradients

(b) Baseline ResNet evaluated with gradients of *different* models

Figure 3: Discriminative pixel perturbation results (higher is better) on the CIFAR100 dataset (see § 5.2.1). We see that score-matched and gradient-norm regularized models best explain model behaviour in both cases, while the anti-score-matched model performs the worst. This agrees with the hypothesis (stated in § 3) that alignment of implicit density models improves gradient explanations and vice versa.

trained models on their own input-gradients and plot the results in Figure 3a. These results indicate that the input-gradients of score-matched and gradient-norm regularized models are better equipped to identify least relevant pixels in this model. However, it is difficult to completely disentangle the robustness benefits of such score-matched models against improved identification of less relevant pixels through such a plot.

To this end, we conduct a second experiment in Figure 3b, where we use input-gradients obtained from these four trained models to explain the same standard baseline ResNet model. This disentangles the robustness of different models as inputs to the same model is perturbed in all cases. Here also we find that gradients from score-matched and gradient-norm regularized models explain behavior of standard baseline models better than the gradients of the baseline model itself. Together, these tests show that training with score-matching indeed produces input-gradients that quantitatively more explanatory than baseline models.

### 5.2.2 QUALITATIVE GRADIENT VISUALIZATIONS

We visualize the structure of logit-gradients of different models in Figure 4. We observe that gradient-norm regularized model and score-matched model have highly perceptually aligned gradients, when compared to the baseline and anti-score-matched gradients, corroborating the quantitative results.

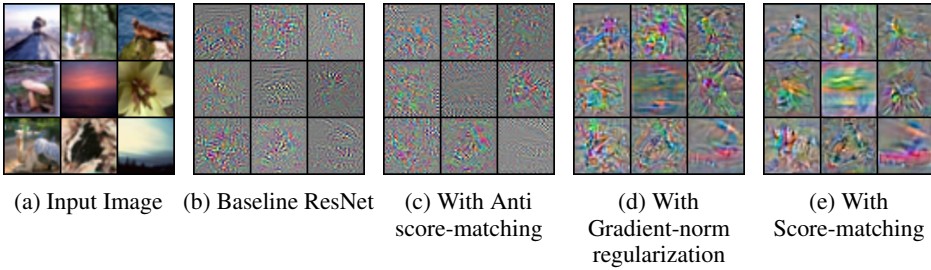

(a) Input Image  (b) Baseline ResNet  (c) With Anti score-matching  (d) With Gradient-norm regularization  (e) With Score-matching

Figure 4: Visualization of input-gradients of different models. We observe that gradients of score-matched and gradient-norm regularized models are more perceptually aligned than the others, with the gradients of the anti-score-matched model being the noisiest. This qualitatively verifies the hypothesis stated in § 3.

## 6 CONCLUSION

In this paper, we investigated the cause for the highly structured and explanatory nature of input-gradients in standard pre-trained models, and showed that alignment of the implicit density model with the ground truth data density is a possible cause. This density modelling interpretation enabled us to view canonical approaches in interpretability such as gradient-based saliency methods, activity maximization and the pixel perturbation test through a density modelling perspective, showing that these capture information relating to the implicit density model, not the underlying discriminative model which we wish to interpret. This calls for a need to re-think the role of these tools in the interpretation of discriminative models. For practitioners, we believe it is best to avoid usage of logit gradient-based tools, for interpretability. If unavoidable, it is recommended to use only gradient-norm regularized or score-matched models, as input-gradients of these models produce more reliable estimates of the gradient of the underlying distribution. As our experiments show, these may be a useful tool even though they are not directly related to the discriminative model.

However, our work still does not answer the question of why pre-trained models may have their implicit density models aligned with ground truth in the first place. One possible reason could be the the presence of an implicit gradient norm regularizer in standard SGD, similar to that shown independently by Barrett & Dherin (2020). Another open question is to understand why gradient-norm regularized models are able to perform implicit density modelling as observed in our experiments in § 5.1.2, which lead to improved gradient explanations.

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

# Appendix

## A   Fooling Gradients is simple

**Observation.** *Assume an arbitrary function $g : \mathbb{R}^D \to \mathbb{R}$. Consider another neural network function given by $\tilde{f}_i(\cdot) = f_i(\cdot) + g(\cdot)$, for $0 \leq i \leq C$, for which we obtain $\nabla_{\mathbf{x}} \tilde{f}_i(\cdot) = \nabla_{\mathbf{x}} f_i(\cdot) + \nabla_{\mathbf{x}} g(\cdot)$. For this, the corresponding loss values and loss-gradients are unchanged, i.e.; $\tilde{\ell}_i(\cdot) = \ell_i(\cdot)$ and $\nabla_{\mathbf{x}} \tilde{\ell}_i(\cdot) = \nabla_{\mathbf{x}} \ell_i(\cdot)$.*

*Proof.* The following expressions relate the loss and neural network function outputs, for the case of cross-entropy loss and usage of the softmax function.

$$\ell_i(\mathbf{x}) = -f_i(\mathbf{x}) + \log\left( \sum_{j=1}^{C} \exp(f_j(\mathbf{x})) \right) \tag{5}$$

$$\nabla_{\mathbf{x}} \ell_i(\mathbf{x}) = -\nabla_{\mathbf{x}} f_i(\mathbf{x}) + \sum_{j=1}^{C} p_j \nabla_{\mathbf{x}} f_j(\mathbf{x}) \tag{6}$$

Upon replacing $f_i$ with $\tilde{f}_i = f_i + g$, the proof follows. $\qquad\square$

### A.1   Manipulating Loss-Gradients

Here, we show how we can also change loss-gradients arbitrarily without significantly changing the loss values themselves. In this case, the trick is to add a high frequency low amplitude sine function to the loss.

**Observation.** *Consider $g(\mathbf{x}) = \epsilon \sin(m\mathbf{x})$, and $\tilde{\ell}_i(\mathbf{x}) = \ell_i(\mathbf{x}) + g(\mathbf{x})$, for $\epsilon, m \geq \mathbb{R}_+$ and $\mathbf{x} \in \mathbb{R}^D$. Then, it is easy to see that $|\tilde{\ell}_i(\mathbf{x}) - \ell_i(\mathbf{x})| \leq \epsilon$, and $\|\nabla_{\mathbf{x}} \tilde{\ell}_i(\mathbf{x}) - \nabla_{\mathbf{x}} \ell_i(\mathbf{x})\|_1 \leq m \times \epsilon \times D$.*

Thus two models with losses differing by some small $\epsilon$ can have gradients differing by $m \times \epsilon \times D$. For $m \to \infty$ and a fixed $\epsilon$, the gradients can diverge significantly. Thus, loss-gradients are also unreliable, as two models with very similar loss landscapes and hence discriminative abilities, can have drastically different loss-gradients.

This simple illustration highlights the fact that gradients of high-dimensional black-box models are not well-behaved in general, and this depends on both the model smoothness and the high-dimensionality of the inputs. Further, loss values and loss-gradients for highly confident samples are close to zero. Thus any external noise added (due to stochastic training, for instance) can easily dominate the loss-gradient terms even when smoothness conditions (small $m$) are enforced.

## B   Score-Matching Approximation

We consider the approximation derived for the estimator of the Hessian trace, which is first derived from Hutchinson's trace estimator Hutchinson (1990). We replace $\log p_\theta(\mathbf{x})$ terms used in the main text with $f(\mathbf{x})$ terms here for clarity. The Taylor series trick for approximating the Hessian-trace is given below.

$$\begin{aligned} \mathbb{E}_{\boldsymbol{v} \sim \mathcal{N}(0, \mathrm{I})} \, \boldsymbol{v}^{\mathsf{T}} \nabla_{\mathbf{x}}^2 f(\mathbf{x}) \boldsymbol{v} &= \frac{1}{\sigma^2} \mathbb{E}_{\boldsymbol{v} \sim \mathcal{N}(0, \sigma^2 \mathrm{I})} \boldsymbol{v}^{\mathsf{T}} \nabla_{\mathbf{x}}^2 f(\mathbf{x}) \boldsymbol{v} \\ &= \frac{2}{\sigma^2} \mathbb{E}_{\boldsymbol{v} \sim \mathcal{N}(0, \sigma^2 \mathrm{I})} \left( f(\mathbf{x} + \boldsymbol{v}) - f(\mathbf{x}) - \nabla_x f(\mathbf{x})^{\mathsf{T}} \boldsymbol{v} + \mathcal{O}(\sigma^3) \right) \end{aligned} \tag{7}$$

As expected, the approximation error vanishes in the limit of small $\sigma$. Let us now consider the finite sample variants of this estimator, with $N$ samples. We shall call this the *Taylor Trace Estimator*.

$$\text{Taylor Trace Estimator (TTE)} = \frac{2}{N\sigma^2} \sum_{i=1}^{N} \big(f(\mathbf{x} + \boldsymbol{v}_i) - f(\mathbf{x})\big) \quad \text{s.t.} \quad \boldsymbol{v}_i \sim \mathcal{N}(0, \sigma^2 \mathbf{I}) \quad (8)$$

We shall henceforth suppress the dependence on $i$ for brevity. For this estimator, we can compute its variance for quadratic functions $f$, where higher-order Taylor expansion terms are zero. We make the following observation.

**Observation.** *For quadratic functions $f$, the variance of the Taylor Trace Estimator is greater than the variance of the Hutchinson estimator by an amount at most equal to $4\sigma^{-2}\|\nabla_{\mathbf{x}} f(\mathbf{x})\|^2$.*

*Proof.*

$$
\begin{aligned}
\text{Var(T.T.E.)} &= \frac{1}{\sigma^4} \mathbb{E}_v \left( \frac{2}{N} \sum_{i=1}^{N} \big(f(\mathbf{x} + \boldsymbol{v}) - f(\mathbf{x})\big) - \mathbb{E}_v \boldsymbol{v}^\mathsf{T} \nabla_{\mathbf{x}}^2 f(\mathbf{x}) \boldsymbol{v} \right)^2 \\
&= \frac{1}{\sigma^4} \mathbb{E}_v \bigg( \frac{2}{N} \sum_{i=1}^{N} \big(f(\mathbf{x} + \boldsymbol{v}) - f(\mathbf{x})\big) - \frac{1}{N} \sum_{i=1}^{N} \boldsymbol{v}^\mathsf{T} \nabla_{\mathbf{x}}^2 f(\mathbf{x}) \boldsymbol{v} \\
&\quad + \frac{1}{N} \sum_{i=1}^{N} \boldsymbol{v}^\mathsf{T} \nabla_{\mathbf{x}}^2 f(\mathbf{x}) \boldsymbol{v} - \mathbb{E}_v \boldsymbol{v}^\mathsf{T} \nabla_{\mathbf{x}}^2 f(\mathbf{x}) \boldsymbol{v} \bigg)^2 \\
&= \frac{1}{\sigma^4} \mathbb{E}_v \left( \frac{2}{N} \sum_{i=1}^{N} \big(f(\mathbf{x} + \boldsymbol{v}) - f(\mathbf{x})\big) - \frac{1}{N} \sum_{i=1}^{N} \boldsymbol{v}^\mathsf{T} \nabla_{\mathbf{x}}^2 f(\mathbf{x}) \boldsymbol{v} \right)^2 \\
&\quad + \frac{1}{\sigma^4} \mathbb{E}_v \left( \frac{1}{N} \sum_{i=1}^{N} \boldsymbol{v}^\mathsf{T} \nabla_{\mathbf{x}}^2 f(\mathbf{x}) \boldsymbol{v} - \mathbb{E}_v \boldsymbol{v}^\mathsf{T} \nabla_{\mathbf{x}}^2 f(\mathbf{x}) \boldsymbol{v} \right)^2
\end{aligned}
$$

Thus we have decomposed the variance of the overall estimator into two terms: the first captures the variance of the Taylor approximation, and the second captures the variance of the Hutchinson estimator.

Considering only the first term, i.e.; the variance of the Taylor approximation, we have:

$$
\begin{aligned}
\frac{1}{N\sigma^4} \mathbb{E}_v \left( 2 \sum_{i=1}^{N} \big(f(\mathbf{x} + \boldsymbol{v}) - f(\mathbf{x})\big) - \sum_{i=1}^{N} \boldsymbol{v}^\mathsf{T} \nabla_{\mathbf{x}}^2 f(\mathbf{x}) \boldsymbol{v} \right)^2 &= \frac{4}{N\sigma^4} \mathbb{E}_v \left( \sum_{i=1}^{N} \nabla_{\mathbf{x}} f(\mathbf{x})^T \boldsymbol{v} \right)^2 \\
&\leq \frac{4}{\sigma^4} \|\nabla_{\mathbf{x}} f(\mathbf{x})\|^2 \mathbb{E}_v \|\boldsymbol{v}\|^2 \\
&= 4\sigma^{-2} \|\nabla_{\mathbf{x}} f(\mathbf{x})\|^2
\end{aligned}
$$

The intermediate steps involve expanding the summation, noticing that pairwise terms cancel, and applying the Cauchy-Schwartz inequality. $\qquad\square$

Thus we have a trade-off: a large $\sigma$ results in lower estimator variance but a large Taylor approximation error, whereas the opposite is true for small $\sigma$. However for functions with small gradient norm, both the estimator variance and Taylor approximation error is small for small $\sigma$. We note that when applied to score-matching Hyvärinen (2005), the gradient norm of the function is also minimized. This implies that in practice, the gradient norm of the function is likely to be low, thus resulting in a small estimator variance even for small $\sigma$. The variance of the Hutchinson estimator is given below for reference Hutchinson (1990); Avron & Toledo (2011):

$$\text{Var(Hutchinson)} = \frac{2}{N} \|\nabla_{\mathbf{x}}^2 f(\mathbf{x})\|_F^2$$

## C EVALUATING EFFECT OF SCORE-MATCHING ON GRADIENT EXPLANATIONS (ON CIFAR10)

We repeat the pixel perturbation experiments on the CIFAR10 dataset and we observe similar qualitative trends. In both cases, we observe that score-matched and gradient norm regularized models have more explanatory gradients, while anti-score-matched model contains the least explanatory gradients. We also present visualization results of input-gradients of various models for reference.

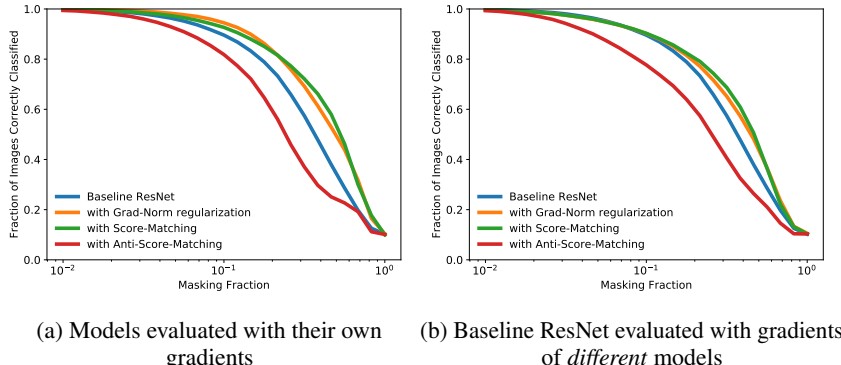

(a) Models evaluated with their own gradients

(b) Baseline ResNet evaluated with gradients of *different* models

Figure 5: Discriminative pixel perturbation results (higher is better) on the CIFAR10 dataset (see § 5.2.1). We see that score-matched and gradient-norm regularized models best explain model behaviour in both cases, while the anti-score-matched model performs the worst. This agrees with the hypothesis (stated in § 3) that alignment of implicit density models improves gradient explanations and vice versa.

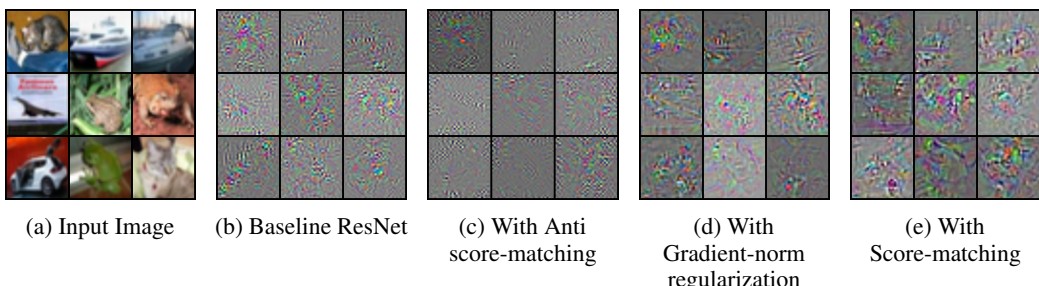

(a) Input Image    (b) Baseline ResNet    (c) With Anti score-matching    (d) With Gradient-norm regularization    (e) With Score-matching

Figure 6: Visualization of input-gradients of different models. We observe that gradients of score-matched and gradient-norm regularized models are more perceptually aligned than the others, with the gradients of the anti-score-matched model being the noisiest. This qualitatively verifies the hypothesis stated in § 3.

## D HYPER-PARAMETER SWEEP ON SCORE-MATCHED TRAINING

We present results on a hyper-parameter sweep on the $\lambda$ and $\mu$ parameters of score-matching, where we provide both test-set accuracy on CIFAR100 and the corresponding GAN-test scores. We find upon performing a hyper-parameter sweep that $\lambda = 1e - 5$ and $\mu = 1e - 3$ seems to perform the best, whereas in the main paper we present results for $\lambda = 1e - 3$ and $\mu = 1e - 4$. It is possible that changing the training schedule by increasing the number of epochs or learning rate may further improve these results, but we did not explore that here.

| $\lambda\downarrow/\mu\rightarrow$ | $1e-2$ | $1e-3$ | $1e-4$ | $1e-5$ |
|---|---|---|---|---|
| $1e-2$ | 48.68%/51.60% | 64.57%/58.90% | 64.75%/76.46% | 9.08%/0.97% |
| $1e-3$ | 64.64%/56.78% | 71.37%/40.72% | 72.34%/73.39% | 34.46%/3.3% |
| $1e-4$ | 69.85%/41.30% | 73.97%/72.07% | 75.65%/79.39% | 72.97%/61.52% |
| $1e-5$ | 73.29%/68.94% | **75.37**%/**85.64**% | 76.40%/63.96% | 74.80%/78.41% |
| $1e-6$ | 75.43%/82.81% | 75.90%/66.11% | 76.77%/65.91% | 75.91%/65.52% |

Table 2: Results of a hyper-parameter sweep on $\lambda$ and $\mu$. The numbers presented are in the format (accuracy % / GAN-test % )

