# OpenReview forum: "Rethinking the Role of Gradient-based Attribution Methods for Model Interpretability"
_ICLR.cc/2021/Conference — ICLR 2021 Oral_

### Official Review · AnonReviewer2 · 2020-10-24
**Needs to be reworked to have a clear message and support evidence**

**Rating:** 5
**Confidence:** 4

**Review:**

This paper investigates the utility of interpreting deep classification models using the gradients of the output logits w.r.t the inputs, a common practice that is also potentially misleading.

The hypothesis stated for the paper is "input-gradients are highly structured because this implicit density model is aligned with the ‘ground truth’ class-conditional data distribution?"

For the observation in section 2 (also maybe number blocks like that similar to lemmas and hypotheses to make it easier to refer to them) if g is 0 then this is trivially true and not saying anything. I think I know what you want to say but this formalism is not adding any clarity. I think adding some constraints to g or simply calling it a variable which takes on specific values. It is not clear why it needs to be conditioned on x either.

3.3 Stabilized score matching: It seems multiple published methods did not help you prevent collapse of the Hessian-trace, but your heuristic did. Is this a common trick or a novel contribution? It would be nice to indicate it one way or another, if it solves a real problem that previous score-matching approaches fail to solve. Furthermore, it would be important to know the sensitivity of your approach to the choice of this hyperparameter.

The importance of section 4.1 is not clear to me… it appears that the authors believe activity maximization is a biased explainability measure and therefore should not be used if one accepts their framework. Their intention for this paragraph should be more explicit.

Section 4.2 draws a tight parallel between the pixel perturbation test and their density ratio test, demonstrating that the pixel perturbation test captures the sensitivity of the implicit density model and not the discriminative model. They therefore suggest this test always be done while tracking changes in classification output, which is a nice takeaway.

In section 4.3, the authors draw a parallel between their score-matching objective and adversarial training, although they state that “score matching imposes a stronger constraint”. I am not an expert on these topics, but I think these kinds of more speculative observations should be moved to the discussion in general.

In section 5, the authors introduce their experimental setup. They used a baseline model ResNet-18 model with 78.01% accuracy, and compared it with their regularized model that only achieved 72.20% accuracy (5.8% drop). The authors weight the entire regularization term with a single lambda =1e-3. This raises a few important questions.
First, the strength of the stability regularizer and the score matching terms should likely be decoupled to achieve maximum classification performance which is crucial in practice. The difficulty of tuning these hyperparamaters are also extremely important for us to understand the utility of the Author’s proposed approach. It would be good for us to see the results of a proper hyperparamater search over the weighting of the score-matching and stability regularization term independently. Ideally, the performance drop observed can be reduced or eliminated.

Second, presumably, if the score matching loss itself hurts classification performance, then intuitively the intuitions built early in the paper cannot be correct: if the aligned density functions p_theta(X) and p_data(X) do can not arise from logits that produce the optimal classifier, then the saliency maps produced by this method cannot be used to diagnose model correctness (as the practitioners who utilize saliency maps would hope). Since the score matching loss is only accurate up to some constant, perhaps this is the source of the issue, but we cannot conclude one way or the other from the data provided.

Furthermore, few details are given for the training setup: how long was each model trained for, was early stopping employed, were multiple seeds evaluated, are we convinced that all models converged? These are important for doing the relevant comparisons, and many of the later results are hard to interpret in the absence of hyperparamater tuning or these experimental details.

Anti-score matching is a great baseline experiment, but the use of the threshold seems arbitrary. This is doublyThis doubly true because the lambda of the regularization is smaller (1e-4). Why this discrepancy? It makes it harder for us to compare the results. How does this model perform as that threshold is varied? If one does a hyperparamater search with a fixed threshold, one would find the best performing anti-score matched model, which would potentially be easier to interpret. Crucially, this approach seems to actually outperform the score-matching model and underperform the baseline, implying that either the lower lambda or the anti-score constraint improved the performance of the classifier, but we cannot know which.

A similar comment can be made about the gradient-norm model RE: hyperparameter searches.

The Density Ratio experiments are excellent, but the y axis is hard to read.

For the Sample Quality experiments: why would the gradient norm model outperform score matching? I am not convinced by the speculation given in the paper. It might be the case that the score-matched models were not converged, highlighting the importance of improving or simply explaining the experimental setup as I mentioned earlier.

Finally, the Gradient Visualization experiments are very unclear. Intuitively, it might be the case that a small portion of the image is enough to explain the class of the data, which is what saliency maps are typically used for. The gradient norm approach and your score matching approach appear to perform almost identically, and it isn’t clear how much better they perform in a practical sense. It would be nice to have a more convincing demonstration of examples where the model obviously classifieds an image using the appropriate information present in using your method but not the baseline. As it stands, your results appear to be largely due to the fact that the gradients are smoother when using score matching or gradient-norm regularization.

In summary, I really like the approach and theory presented in the paper, and tackles an important issue with broad relevance to the field, but the experimental results as they stand are not sufficient to convince me that this approach works in practice.

Typo Section 4: "show how these can interpreted from"

---

> ### Author Response · Authors · 2020-11-12
> **Response - Part 1**
>
> We thank the reviewer for their detailed review. Our response to the questions are below.
>
> ---
>
> " For the observation in section 2 (also maybe number blocks like that similar to lemmas and hypotheses to make it easier to refer to them) if g is 0  then this is trivially true and not saying anything..."
>
> We're sorry that this observation was unclear. Our claim here is that one can add any *arbitrary scalar-valued function* g(x) to the logits of neural network and this leaves the softmax outputs unchanged. One can think of this function g(x) as an arbitrary neural network with a single scalar output and arbitrary input-gradients. This implies that the overall logit gradients are the sum of the "original" logit gradients and the gradients of this arbitrary scalar-valued function, thus making overall logit-gradients arbitrary. Note that this would not hold if we used constant scalar variables instead of scalar-valued functions (as the derivative of a constant is zero), thus necessitating the conditioning on "x".
>
> ---
>
> " Stabilized score matching: It seems multiple published methods did not help you prevent collapse of the Hessian-trace, but your heuristic did. Is  this a common trick or a novel contribution? ... "
>
> This trick is a novel contribution as far as we are aware. We are also the first to apply score-matching to large-scale discriminative neural networks, different from the settings and architectures considered in other works on score-matching. We will make this clearer in the draft. For setting the value of this hyperparameter, we use the smallest possible value that prevents a training collapse due to the instability of score-matching.
>
> ---
>
> " The importance of section 4.1 is not clear to me… "
>
> Our main takeaway from this section is indicated in the last line: "given the fact that they (activity maximization algorithms) draw samples from the implicit density model, their utility in interpreting discriminative models may be limited"
>
> ---
>
> " This raises a few important questions. First, the strength of the stability regularizer and the score matching terms should likely be decoupled to achieve maximum classification performance which is crucial in practice. "
>
> Our approach in this paper was to apply the largest possible lambda to obtain the strongest generative models for the purpose of validating our hypothesis, while achieving competitive discriminative performance. For the stability regularizer, we chose the smallest possible value that avoids training collapse. We perform a broad search in steps of multiples of 10 for this case. It is indeed possible that a finer search would provide a better compromise, however we did not investigate this. We will present the results of a hyper-parameter sweep in a later comment.
>
> ---
>
> " Second, presumably, if the score matching loss itself hurts classification performance, then intuitively the intuitions built early in the paper cannot be correct..."
>
> Our overall message is that usage of gradient-based saliency maps should be avoided for diagnosing model correctness in the first place, as they encode information regarding the implicit density model and not the discriminative model which we wish to diagnose. Our experiments with score-matching are only a tool used to validate this hypothesis that saliency map quality depends on this implicit density model. Our experimental results in Figure 3 show that gradients from score-matched models perform the best, followed by the baseline and the anti-score-matched models, which is in accordance with our hypothesis. We are not sure if this answers the question, so please do respond for further clarification.
>
> ---
>
> " Furthermore, few details are given for the training setup: how long was each model trained for, was early stopping employed, were multiple seeds evaluated, are we convinced that all models converged? "
>
> We train all models for 200 epochs, with an initial learning rate of 1e-1 which is decayed by 10 at 100 and 150 epochs respectively, which is the standard training setup for ResNets on CIFAR100. We did not observe large variation across seeds. We also observed performance to have plateaued at the end of training, making it likely that the models have converged.

---

> > ### Author Response · Authors · 2020-11-12
> > **Response - Part 2**
> >
> > We continue the response to AnonReviewer2 in this comment.
> >
> > ---
> >
> > "The Density Ratio experiments are excellent, but the y axis is hard to read. "
> >
> > Thank you for this suggestion, we will change this in an update of our draft.
> >
> > ---
> >
> > "For the Sample Quality experiments: why would the gradient norm model outperform score matching? ..."
> >
> > We agree that this phenomenon is unusual, and we leave the investigation of this to future work, as specified in the conclusion. It is certainlypossible that score-matching models underperform due to suboptimal regularization, but we believe that the models have converged because of the performance plateau we observe, and any suboptimality in generative modelling may be due to instability of the score-matching framework itself. Please note that this still verifies our hypothesis: both gradient norm regularized models and score-matched models have better implicit density models than the baseline model (as shown in Section 5.1), which leads to improved saliency maps (as shown in Section 5.2).
> >
> > ---
> >
> > " It would be nice to have a more convincing demonstration of examples where the model obviously classifieds an image using the appropriate information present in using your method but not the baseline. As it stands, your results appear to be largely due to the fact that the gradients are smoother when using score matching or gradient-norm regularization. "
> >
> > We evaluate these saliency maps quantitatively in Figure 3 using the pixel perturbation test, which exactly measures this effect. Figure 3 shows that score-matched and gradient-regularized models indeed outperform baseline models in identifying "important" pixels according to this test.
> >
> > ---
> >
> > " In summary, I really like the approach and theory presented in the paper, and tackles an important issue with broad relevance to the field, but the experimental results as they stand are not sufficient to convince me that this approach works in practice. "
> >
> > We thank you for your positive comments regarding the theory presented in this paper, and we hope to answer any more questions regarding the relevance of the experimental results, or clear any confusion that may remain.

---

> > ### Author Response · Authors · 2020-11-22
> > **Results of hyper-parameter sweep**
> >
> > "The difficulty of tuning these hyperparamaters are also extremely important for us to understand the utility of the Author’s proposed approach. It would be good for us to see the results of a proper hyperparamater search over the weighting of the score-matching and stability regularization term independently. Ideally, the performance drop observed can be reduced or eliminated."
> >
> > We present results on a hyper-parameter sweep on the $\lambda$ and $\mu$ parameters of score-matching, where we provide both test-set accuracy on CIFAR100 and the corresponding GAN-test scores. We find upon performing a hyper-parameter sweep that $\lambda=1e-5$ and $\mu =1e-3$ seems to perform the best, whereas in the paper we present results for $\lambda=1e-3$ and $\mu=1e-4$ (mistakenly indicated as $\mu=1e-3$ in the draft). It is possible that changing the training schedule by increasing the number of epochs or learning rate may further improve these results, but we did not explore that here. However please note that this does not change the main findings of the paper, and only strengthens it by showing that more accurate score-matched models can be obtained. We will add these results in the supplementary section of the paper and include a short discussion in the main paper.
> >
> > | $\lambda \downarrow / \mu \rightarrow$ | $1e-2$ | $1e-3$ | $1e-4$ | $1e-5$ |
> > | --- | --- | ---| --- | --- |
> > | $1e-2$ | 48.68 \% / 51.60\%  | 64.57\% / 58.90\%| 64.75\% / 76.46\% | 9.08 \% / 0.97\% |
> > | $1e-3$ | 64.64\% / 56.78\% | 71.37\% / 40.72\% | 72.34\% / 73.39\% | 34.46\% / 3.3\% |
> > | $1e-4$ | 69.85\% / 41.30\% | 73.97\% / 72.07\% | 75.65\% / 79.39\% | 72.97\% / 61.52\% |
> > | $1e-5$ | 73.29\% / 68.94\% | **75.37\% / 85.64\%** | 76.40\% / 63.96\% | 74.80\% / 78.41\% |
> > | $1e-6$ | 75.43\% / 82.81\% | 75.90\% / 66.11\% | 76.77\% / 65.91\% | 75.91\% / 65.52\% |
> > *Table*: Results of hyper-parameter sweep on score-matching. The numbers presented are in the format **(accuracy % / GAN-test %)**.

---

### Official Review · AnonReviewer4 · 2020-10-28
**Paper provides interesting perspective on input-gradients and their explanatory power**

**Rating:** 7
**Confidence:** 4

**Review:**

In this work the authors explore the link between the explanatory power of input-gradients and the alignment between the "implicit density model" of the softmax-based deep model and the "ground truth" class-conditional data distribution. The authors propose using score-matching method to create models with varying degrees of alignment. The paper is full of interesting insights and ideas, such as soft-max shift invariance property and trivial input-gradient perturbation, connections between score-matching and adversarial training, and others. However, in the end, the paper's impact on how ML engineers will use interpretability tools is in my opinion limited. The authors successfully introduce some interesting heuristics to make the training of score-matching models more scalable and stable. However even with those heuristics ,gradient-norm regularized models are comparable if not superior in the 3 evaluations presented by the authors: GAN-test score, pixel perturbation results, and perceptual alignment of input-gradients. The authors provide enough evidence to validate the main hypothesis of the paper. There are no inconsistencies or errors that I can see in the paper to the best of my knowledge. The paper is clearly written and well structured. To improve the paper, the authors could add some comments expanding on the practical impact that this results will have on the work of ML engineers who use input-gradients as a tool to improve model accuracy.

---

> ### Author Response · Authors · 2020-11-12
> **Regarding Practical Implications of Our Work**
>
> Thank you for the positive and encouraging review of the paper. We respond below to the concern regarding practical implications.
>
> " To improve the paper, the authors could add some comments expanding on the practical impact that this results will have on the work of ML engineers who use input-gradients as a tool to improve model accuracy. "
>
> Thanks for this suggestion. We believe that input-gradient based tools must be avoided when interpreting discriminative models in general. However, if unavoidable, then it is recommended to use input-gradients only on gradient regularized or score-matched models as these provide more reliable estimates of the log-gradients of the underlying data distribution, which may be a useful tool, although they are not directly related to the underlying discriminative model which we wish to interpret. We will add these to an updated version of our draft.

---

### Official Review · AnonReviewer3 · 2020-10-29
**Insightful paper with far reaching implications for gradient-based visualization methods**

**Rating:** 9
**Confidence:** 4

**Review:**

Summary\
The key message of this paper is that input-gradients (gradient of the logit wrt to input) or loss-gradients are/might be unrelated to the discriminative capabilities of a DNN. The input-gradient is a key primitive in several interpretability and visualization methods. Until now, it has been taken as a given that these gradients reveal 'why' or what parts of the inputs the model is sensitive to. However, this paper questions this reasoning and says that if the input-gradients can be easily manipulated without changing the generalization ability of the model, then does the input-gradient really contain discriminative signals?

To test their hypothesis, the paper re-interprets the input-gradient as class-conditional generative model using the score matching view. The paper then develops a 'regularizer' that is called a taylor trace estimator that requires less backward passes than the hutchinson estimator, which when added to the model objective can modulate how 'generative' the model is. With this regularizer, the papers tests the hypothesis that improving the implicit density model also improves input-gradient interpretability. The paper tests this through experiments on an image dataset and finds that this is the case.

Significance\
This work has far reaching significance for the field of visual interpretability of DNNs. It suggests that reading into these input-gradients might be akin to reading tea leaves. The key insight in this work is simple and the demonstration is quite powerful in my opinion. The argument in this paper seems obvious in hindsight, but that is exactly why the paper is a significant one. I have several additional questions later in my review, but this work is important and suggests that insights based on input-gradients might be spurious.

Clarity\
Overall, the paper is relatively clear and easy to read. Several of the key experiments are well-justified.


Originality\
The insight in this paper via the score-matching perspective is new in the interpretability domain. The claim that input-gradients can be easily manipulated is not new, but the general insight in this work is new and important. Overall, this paper opens up several questions about what input-gradients really convey.


Taylor Trace Estimator\
I am confused about the derivation of this estimator and I am probably missing something, so can you walk me through this? Let's say the Taylor series expansion around a point x is:

$f(y) = f(x) + \nabla f(x)^\top(y-x) + \frac{1}{2}(y-x)^\top\nabla^2f(x)(y-x) + \mathcal{O}(\left\Vert y-x\right\Vert^3)$
now if solve for $(y-x)^\top\nabla^2f(x)(y-x) $, we get:
$\frac{1}{2}(y-x)^\top\nabla^2f(x)(y-x) = f(y) - f(x) -  \nabla f(x)^\top(y-x) -  \mathcal{O}(\left\Vert y-x\right\Vert^3)$
$v= y-x$ in your notation, and it is a zero mean gaussian, so we get: $\mathbb{E}[\nabla f(x)^\top v] = 0$, which leads to the approximation that you get. However, where did the $\frac{1}{\sigma^2}$ outside the expectation come from in the final form?

The point on Adversarial Training\
In section 4, this paper notes that recent work has shown that when a model is trained with explanation penalization, it results in more 'interpretable' gradients. This connection was made more formal in recent work (https://arxiv.org/abs/1810.06583.pdf)  that notes that training models while penalizing their integrated gradients explanations is equivalent (Thm 5.1 in that paper, for the right loss function and some other assumptions) to $\ell-\infty$ adversarial training.

Other Feedback and Questions
-  The current section 4 is really a discussion/implications section. I suggest the authors call it that. It should also likely come after section 5 since that is where the experimental results are. From reading section 4, am I right to conclude that the paper is also suggesting that activation maximization, pixel perturbation, and the results of adversarial training say more about the implicit generative model than discriminative information for a DNN? That is, I should also not take the results of activation maximization as explaining to me what a neuron that learned?

- Do these results extend to methods that post-process or use the input-gradients as a primitive? For example, smoothgrad adds noise to the input and takes an average of the corresponding input gradients. Integrated gradients can be seen as as sum of interpolated input-gradients along an all-zeroes input to an input of interest. Should I also take it from these results that Smoothgrad and integrated gradients don't indicate discriminative behavior as well?

- For example, consider grad-cam, which looks at the output of a convolutional layer in computing a sensitivity map as opposed to the logits, does your analysis apply to that case too? I think it probably doesn't, unless one can also view the output of convolutional filters are implicit density models as well.

- Does the analysis in section 2 apply to the probability output from softmax? I.e., can the 'probability-gradients' also be arbitrarily manipulated?

Overall, this work raises several important questions like why trained models have implicit density models that are aligned with the inputs in the first place. This question seems key to tying up the remaining loose ends in this work. This said, this paper is still a thought-provoking one and a useful one for the literature on DNN interpretability.

---

> ### Author Response · Authors · 2020-11-12
> **Clarifications**
>
> Thank you for the positive and encouraging review of the paper! We are glad that you liked it and found the theory insightful. We respond to the questions below.
>
> ---
>
> Taylor Trace Estimator: This is a subtle point, but please note that RHS of equation 3 considers "v" drawn from a normal distribution scaled by $\sigma$. We hence compensate for this by dividing by $\sigma^2$ outside the expectation.
>
> ---
>
> Adversarial Training: Thank you for providing this reference. Broadly, this paper makes the connection for a special variant of IG where the baseline is chosen to be $\epsilon$ close to the original input, which maximizes the IG norm. While this variant of IG has not been used for interpretability previously, it still makes an interesting connection, and we will add discussion relating to this to our paper.
>
> ---
>
> " The current section 4 is really a discussion/implications section. I suggest the authors call it that. It should also likely come after section 5 since that is where the experimental results are. "
>
> Thank you for these suggestions, we will incorporate these in our draft.
>
> ---
>
> " From reading section 4, am I right to conclude that the paper is also suggesting that activation maximization, pixel perturbation, and the results of adversarial training say more about the implicit generative model than discriminative information for a DNN? That is, I should also not take the results of activation maximization as explaining to me what a neuron that learned?"
>
> This is indeed correct for activity maximization and pixel perturbation. For the case of adversarial training we only point similarities to implicit density modelling and not a precise connection. Regarding activation maximization of logits, we claim that the quality of the resultingimages is not indicative of the discriminative performance, but to that of the implicit density model. This implies that in principle, it is possible to obtain high quality activation maximization images for both highly performant discriminative networks as well as for models with poor discriminative performance. While we do not make a similar claim for activation maximization of intermediate neurons, as they do not have an associated density model, we believe a similar trend may hold in this case as well.
>
> ---
>
> " Do these results extend to methods that post-process or use the input-gradients as a primitive? For example, smoothgrad adds noise to the input and takes an average of the corresponding input gradients. Integrated gradients can be seen as as sum of interpolated input-gradients along an all-zeroes input to an input of interest. Should I also take it from these results that Smoothgrad and integrated gradients don't indicate discriminative behavior as well?"
>
> This is indeed true. Technically, if logits-gradients are used for the computation of smoothgrad or integrated gradients, as is done typically, then these saliency maps aggregate the score-function gradients at different points in the input space. For smoothgrad, we can also interpret this map as the log gradients of a modified density model obtained by "smoothening" the original implicit density model. A related interpretation can be made for integrated gradients.
>
> ---
>
> " For example, consider grad-cam, which looks at the output of a convolutional layer in computing a sensitivity map as opposed to the logits, does your analysis apply to that case too? I think it probably doesn't, unless one can also view the output of convolutional filters are implicit density models as well."
>
> As GradCAM uses logit-gradients to aggregate the convolutional map, we believe that this analysis holds in this case as well.
>
> ---
>
> " Does the analysis in section 2 apply to the probability output from softmax? I.e., can the 'probability-gradients' also be arbitrarily manipulated?"
>
> This is indeed possible. We present a way to manipulate post-softmax loss-gradients in the appendix B.1 by adding a high frequency small amplitude noise to the loss value.

---

> > ### Comment · AnonReviewer3 · 2020-11-23
> > **More Questions**
> >
> > Thanks to the authors for clarifying some of my confusion and providing answer. After reading these responses and digesting the work, I now some additional questions.
> >
> > ### When is a generative model encoded within a discriminative one?
> > The crux of the argument in this paper seems to be that logits are class conditional generative models. The authors allude to the 'score-matching' literature to justify this view.  However, under what assumptions can we view the logits this way? For example, for a binary logistic model, I don't think this is the case.  Is it only for a softmax-based classifier? Specifically, can a softmax logistic multi-class classifier be interpreted in a similar fashion? Section 3 of https://arxiv.org/pdf/1912.03263.pdf is useful, but not quite clear on what assumptions need to be in place for the EBM view to be valid. What if we replace the final layer of say a resnet/deep NN with a single unit and minimize the l2 regression loss. Will this also fit under the view here? In general, I am trying to understand when we can say an output unit encodes a generative model.
> >
> > A basic follow-up question on the TTE: so can this estimate only be used for functions? Let's say I form a random matrix A (k  by k), and I want to use the TTE for this setting, how will I do that? In the case of hutchinson, I just do: v'Av where v is rademacher or gaussian. What is the equivalent for TTE? I know this defeats the purpose of the estimator in some sense since if one could form the matrix, then there is no need to approximate the trace, one could just compute it. I just wonder if I am missing something.

---

> > > ### Author Response · Authors · 2020-11-23
> > > **Response**
> > >
> > > Great questions!
> > >
> > > 1) A small clarification: our justification for the density modelling view arises from these two papers [1,2]. From [1], we also have a similar interpretation for binary logistic classifiers, and by extension, to logistic multi-way classifiers. For a binary classifier, the singular "logit" encodes the log ratio of class conditional probabilities, i.e., $p(x|y=0) / p(x|y=1)$. To see why, note that from Bayes rule, $p(y=1|x) = \frac{1}{1 + (p(x|y=0) / p(x|y=1))}$, and the ratio is encoded by $\exp(-f(x))$, meaning $f(x) = \log p(x|y=1) - \log p(x|y=0)$. For a logistic multi-way classifier, we would similarly have logits encoding $p(x|y=c) / p(x|y \neq c)$. We are not aware of similar interpretations for regression losses. In general, whenever we can re-interpret an output non-linearity as an instance of the Bayes rule, we have such a density modelling interpretation.
> > >
> > > [1]: Bridle, John S. "Probabilistic interpretation of feedforward classification network outputs, with relationships to statistical pattern recognition." Neurocomputing 1990
> > >
> > > [2]: Grathwohl, Will, et al. "Your classifier is secretly an energy based model and you should treat it like one.", ICLR 2020
> > >
> > > 2) Indeed, the TTE can be seen as a trace estimator for Hessian matrices only. Assuming we know (or can construct) the function $f(x)$ whose Hessian trace we wish to estimate, we can apply TTE.

---

### Official Review · AnonReviewer1 · 2020-11-03
**Insightful theory grounded dissection of gradient-based explainability methods**

**Rating:** 9
**Confidence:** 4

**Review:**

This paper examines gradient-based attribution methods that have been proposed in the explainability literature from a theoretical perspective motivated by a recent observation in energy-based generative models.
First, the authors point out a general weakness of gradient-based attribution that derives from the fact that input-gradients do not provide well-defined explanations, since the shift-invariance of the softmax output makes them arbitrary.
The authors then propose that the reason for the success of gradient-based attribution models can be explained by the fact that discriminative models "contain an implicit" class-conditional density model (the mentioned recent observation about energy-based generative models).
They then go on to elaborate on this idea showing how aligning the implicit class-conditional generative model to the "true" generative model of the data would help provide relates to gradient-based attribution, how the alignment can be efficiently promoted with a novel implementation of score-matching, and how this mechanism can be practically realized as regularization costs.
The authors then carry out empirical studies that convincingly confirm the prediction of their theoretical ideas. First, they show that samples generated with score-matching and the proposed gradient-norm regularization are better in the sense of being less noisy and in terms of their discriminative accuracy via a trained discriminative model as proposed by the "GAN-test approach".
Finally, they show that the quality of gradient-based explanations are better according to a discriminative version of the pixel perturbation test, a method to evaluate gradient explanations by perturbing pixels ranked in increasing order of relevance.
In conclusion, this paper establishes very interesting fundamental theoretical connections between discriminative models, energy-based generative models, and gradient-based explanations, uses this theoretical framework to explain how gradient-based explanation are overcoming the softmax shift-invariance problem (also pointed out in this paper), and introduces practical training procedures to take advantage of the gained theoretical insights to generate better explanations, which are also empirically verified in simulations.

---

> ### Author Response · Authors · 2020-11-12
> **Thank you!**
>
> We would like to thank the reviewer for their positive assessment of the paper, and for the encouraging feedback.

---

### Public Comment · ~Tianyu_Pang1 · 2020-11-16
**Good paper, related to a prior work titled "Finite-difference Score Matching"**

We understand the main purpose of the paper is to rethink how the logits gradient reflects the interpretability in a discriminative model. This is very interesting.

We find that Eq. (3) in this paper applies a finite-difference (FD) method to approximate the Hessian trace in score matching (SM). Actually, there is a previous work on finite-difference score matching [1], which provides a better FD approximation (i.e., less biased when sampling) and faster implementation (i.e., parallelizing the FD calculations) for the SM methods. We think [1] may be helpful for the proposed method in this paper.

Reference

[1] Efficient Learning of Generative Models via Finite-Difference Score Matching, NeurIPS 2020.

Paper: https://proceedings.neurips.cc/paper/2020/file/de6b1cf3fb0a3aa1244d30f7b8c29c41-Paper.pdf

Code: https://github.com/taufikxu/FD-ScoreMatching

---

> ### Author Response · Authors · 2020-11-18
> **Thank you for the reference**
>
> Hi Tianyu,
>
> Thanks for pointing us to your paper! There are indeed similarities between the finite difference Score-matching approach and our approach for speeding up Hessian trace computation. Most notably, we use uncentered finite differences as opposed to your method which uses centered differences, and hence may have lower variance. We will add a note about this in an update to our draft.

---

### Author Response · Authors · 2020-11-22
**Update to the Draft**

A note on the changes made to the draft.

1) Included discussion of Finite-difference Score Matching (ref. comment by Tianyu Pang). Broad difference between the two is the use of centered vs uncentered finite differences, and the application to discriminative neural networks in our case, as opposed to noise contrastive networks or denoising score matching.
2) Included hyper-parameter sweep experiments in the supplementary material, as suggested by Reviewer2.
3) Included references and suggestions by Reviewer3. We renamed Section 4 which was previously called "Interpretability through the lens of density modelling" to "Implications of the density modelling viewpoint", but did not change its location as this section introduces discriminative pixel perturbation test, which is a pre-requisite for Section 5.
4) Included discussion suggested by Reviewer4 in the conclusion section.

---

### Decision · Program_Chairs · 2021-01-07
**Final Decision**

**Decision:**

Accept (Oral)

**Comment:**

This paper studies why input gradients can give meaningful feature attributions even though they can be changed arbitrarily without affecting the prediction. The claim in this paper is that "the learned logits in fact represent class conditional probabilities and hence input gradients given meaningful feature attributions". The main concern is that this claim is verified very indirectly, by adding a regularization term that promotes logits learning class conditional probabilities and observing that input gradient quality also improves. Nevertheless, there are interesting insights in the paper and the questions it asks are very timely and important, and overall, it could have a significant impact on further research in this area.